# Participation and Inclusion of Children and Youth with Disabilities in Local Communities

**DOI:** 10.3390/ijerph191911893

**Published:** 2022-09-20

**Authors:** Arne H. Eide, Dag Ofstad, Marit Støylen, Emil Hansen, Marikken Høiseth

**Affiliations:** 1SINTEF Digital, Health Research, Forskningsveien 1, N-0314 Oslo, Norway; 2Department of Social Work, Faculty of Social and Educational Sciences, Norwegian University of Science and Technology, N-7491 Trondheim, Norway; 3Norwegian National Advisory Unit on Disability in Children and Youth (NKBFU), Valnesfjord Health Sports Center (VHSS), N-8215 Valnesfjord, Norway; 4Department of Design, Faculty of Architecture and Design, Norwegian University of Science and Technology, N-7491 Trondheim, Norway

**Keywords:** participation, inclusion, local community, children, youth, disabilities

## Abstract

This research aimed at strengthening the evidence base for interventions to promote the participation and inclusion of children and youth with disabilities in their communities. Four selected municipalities in four different regions of Norway participated. Focus group discussions (FGDs) were carried out in all four municipalities with service providers, children and youth with disabilities, and parents of children and youth with disabilities. The FGDs were used to develop a questionnaire to measure participation indicators. A survey was carried out among 186 children and youth, combining (a) random sampling of children and youth without disabilities and (b) invitations to all children and youth with disabilities, as registered by the municipalities. High-level analyses of the FGDs are presented, indicating the barriers and facilitators for participation for children and youth with disabilities, as well as requests for improvement of services to stimulate participation. The categories of family, technology, and volunteering were identified primarily as facilitators of participation, while school environment was primarily identified as a barrier. The survey measured four different aspects of perception of one’s own participation. Data analyses revealed higher level of environmental barriers among children and youth with disabilities, and lower level of actual participation, satisfaction with one’s own participation and one’s own participation compared with that of peers.

## 1. Introduction

For the purpose of this article, children and youth are defined as persons between the ages of 0 and 25 years, in accordance with the definition used by the Norwegian Advisory Unit on Disability in Children and Youth (NKBF). Children and youth with disabilities face specific barriers in accessing their rights and obtaining full participation alongside their non-disabled peers [1,2,3]. 

The responsibility to ensure equal participation and inclusion in society is firmly within the scope of the United Nations Convention on the Rights of Persons with Disabilities (CRPD) [4]. The participation and inclusion of vulnerable groups in society is strongly advocated by the United Nations’ Sustainable Development Goals and the ethos of “leaving no one behind” [5]. Other international and national policies convey the same strong message [6,7,8]. 

While much effort has been put into research on the participation and inclusion of children and youth with disabilities in schools [9], research aimed at strengthening the evidence base for interventions to promote more broadly their participation and inclusion is limited [1]. To develop interventions in local communities that will promote inclusive societies, baseline data on inclusion and participation, as well as innovative approaches to stimulate local communities—where children and youth live their lives and experience barriers—are needed. It is important to bear mind that the local contexts are diverse, as is the evidence about what approaches work, when they work, and where they work. 

This paper aims to contribute to the knowledge base that is necessary for service innovation to strengthen municipalities’ efforts to promote the participation and inclusion of children and youth with disabilities in their daily lives. This study will be carried out by outlining an innovative methodology and using the results and experiences derived from an ongoing research and innovation process in four municipalities in Norway.

Participation and inclusion are concepts with a variety of meanings and understandings. The World Health Organization (WHO) defines participation as “involvement in a life situation” [10]. This rather broad definition is adopted in this paper and includes involvement in schools, families, organized activities such as sports and community activities, unorganized leisure activities, and everyday life. The International Classification of Disability, Functioning and Health (ICF) distinguishes between “capacity” and participation as “performance”, or, put in a different way, what persons “can do” and what they in fact do (“do do”). Participation can be viewed as the outcome of an interaction between health conditions, body functions, activities, environmental factors, and individual factors. The influence of environmental and personal factors determine whether there is a gap between capacity and performance. Children and youth with disabilities have specific functional difficulties that interact with barriers in the environment, influencing their level of participation in different life domains. This is captured in the children and youth version of the ICF [11] (ICF-CY). Indicators of participation may be drawn from the ICF and ICF-CY checklists (https://www.washintongroupdisabiity.com/fileamin/uloads/wg/Docments/Events/11/annex1_-icf_cy_checklist.pdf, accessed on 3 June 2022).

Knight et al. [12] argued that true social inclusion can only be achieved if it is understood as a right and provided with priority at all levels of local authority. Inclusion is understood as the process that leads to participation for all in activities and services (https://www.acedisability.org.au/information-for-providers/inclusion-in-the-community.php, accessed on 3 June 2022). Inclusion and participation are mutually reinforcing. Norwegian municipalities are responsible for providing services to children and adolescents with disabilities, providing an overview of who is entitled to services based on to their health conditions and/or disabilities, establishing a coordinating unit and formalizing individual plans, and striving toward inclusion and participation among children and youth with functional difficulties. While the Norwegian health and welfare system is well developed and rights-based, various problems with resources, competence, and coordination have been identified [13].

Recent literature on participation among children and youth have highlighted the impact of cultural differences [14], the importance of social support by caregivers [15], the need for obtaining childrens’ perspectives on participation [16], and the crucial role of the school environment [17]. These recent examples point in different ways to the interrelationship between the environment and the individual as a key to participation.

Current knowledge within Norwegian municipalities about the participation of children and youth is limited. However, research has indicated that some key mechanisms for achieving inclusion, such as an individual planning and coordinating unit, are not well implemented and vary significantly between municipalities [18,19]. To develop such measures, including new innovative methods, there is need for a deeper understanding of the current situation and of service innovations that involve the municipalities and the target groups in developing contextually relevant strategies for the inclusion of children and youth.

To assist municipalities in improving the quality of services for the target groups, a project known as “Developing a method for strengthening the quality of municipal services to children and youth with disabilities” was funded by the Norwegian Research Council in 2019 as an “innovation project in the public sector”. The project is led by the Norwegian National Advisory Unit on Disabilities in Children & Youth (NKBUF) in collaboration with SINTEF Digital as a research partner, and is based at the Valnesfjord Health Sports Center (VHSS), a rehabilitation institution. The broad mission of this project is to contribute to building competence in all health regions in Norway, by establishing professional networks that will be a driving force in the development of services for children and youth with functional difficulties. To achieve this objective, networks must be developed that involve municipalities, community culture, voluntary sector, regional specialist services, families, and children and youth themselves. An important first step in this process is to generate new knowledge about participation among the target groups and to use this knowledge in a co-creative service design process (defined as a collaborative process of researching, envisaging, and orchestrating experiences that occur over time and multiple touch points https://medium.com/@birchall/what-is-service-design-4f6e0086f501, accessed on 2 June 2022). This process will involve municipalities, families, and children/youth in challenging current practices, leading to improvements in established services and the development of new methods and measures.

While the ICF model provides an understanding of a relational and complex context, it also illustrates that interventions to promote participation have several “entry points”. According to Hoehne et al. [20], interventions targeting environmental barriers to participation have emerged during the last decade. One example is their own study using the pathways and resources for engagement and participation (PREP) that provide coaching to foster problem-solving and self-advocacy skills. Other examples mentioned by Hoehne et al. include an intervention using context-focused therapy for children and another that teaches youth to systematically identify environmental barriers, generate modification strategies, and request accommodations. While such interventions target children and youth to help them manage environmental barriers, no study or intervention was found that targets the environment to promote participation among children and youth.

The ultimate goals of this article are to develop a method for increasing social and daily-life participation of children and youth with reduced functioning and to implement measures to promote the inclusion and participation of children and youth = with functional difficulties in the four participating municipalities. The specific aim for this article is to develop a system for quality measurement, learning, and change in municipal services for the target groups. Subsequently, this research will be followed by an intervention that targets the system level, and more specifically the participating municipalities. This article will present our methodological approach; the high-level results from focus group discussions (FGDs) among parents, service providers, and youths; the development of indicators for participation; and comparative analyses of the levels of participation of children and youth with or without disabilities.

## 2. Materials and Methods

One municipality in each of the four health regions of Norway were included in this study. The four municipalities were selected because of their participation in previous collaborations with the NKBUF; their previous experience of being relatively active in their approaches to children and youth with disabilities; and their motivation to work closely with researchers and innovators to achieve inclusion and participation for the target groups. Further, the municipal management personnel in the four communities were willing to commit to the project.

The municipalities were all of medium size in a Norwegian context, with inhabitants numbering from 12,000 to 30,000. They differed on socio-economic and organizational (municipality) factors and population composition/demography. Accordingly, they represented a variety of conditions for implementing services for children and youth with disabilities.

The key research elements in the project were to generate in-depth knowledge about how participation plays out in daily life; how the barriers, facilitators, and conditions affect participation; and how such factors are linked to municipal services. Focus group discussions are suitable for gaining insights into complicated topics that are related to multi-faceted contexts, behaviors, and motivations. A survey methodology based on in-depth qualitative data can provide quantitative measures of the scope and diversity of participation, comparisons between groups, and conceptual relationships. Qualitative studies combined with a baseline survey and repeated measures provide data that can identify successes and failures in collaboration and coordination within services, optimal or sub-optimal capacities and competencies, under-utilizations of municipal resources, and the potential to ensure inclusive environments. To identify causation and consequences for services and their users, a dialogue process with feedback loops between the services, service users, and researchers is ongoing, together with the research activities within the overall innovation process. Figure 1 illustrates the different steps of the research and innovation in this project.

A foundation for collaboration and co-creation in service development was established through several workshops and bi-lateral meetings with the four municipalities. including both management and service providers (Steps 1 and 2 in Figure 1). Separate contracts were negotiated with the four municipalities to ensure leadership commitment, and an extended project team was established, including a responsible contact person from each municipality.

The third step was to explore the current situation in the municipalities with respect to participation and inclusion. In each municipality, separate focus group discussions were carried out with youth with disabilities, parents of children with disabilities, and relevant service providers who were working operatively with the target group (Step 3). Altogether, this amounted to 12 focus group discussions with a total of 60+ participants. The children and youth had diverse functional difficulties. The diagnoses represented in the study included cerebral palsy, mild cognitive impairments, physical impairments, and severe multifunctional disabilities.

Data from the focus groups were analyzed and reported back to the participants/ municipalities, then used in the following steps. Step 4 comprised the development of indicators of inclusion and participation among the children and youth who were included in the survey (Step 5). Data from the survey were analyzed and reported back to the municipalities. Step 6 will be service design, followed by implementation of changes (Step 7) and design of a template for future use and wider implementation (Step 8). (Steps 6–8 are yet to be implemented and, accordingly, are not presented or discussed in this article. Service design can be defined as “a collaborative process of researching, envisaging and then orchestrating experiences that happen over time and multiple touch points” [21]. The survey may be repeated to identify changes in inclusion/participation among the target groups and used in a feedback process and during a new co-design and implementation cycle.

Focus group discussions normally comprise five to seven persons with similar background and provide insight into different experiences and opinions among individual participants. This method differes from interviews, in that the discussion/dialogue among the participants generates a broader perspective and new insights. In the current project, all focus group discussions were audio-recorded and notes were taken by a researcher. Three of the focus groups had fewer participants (two or three), while the number of participants in the remaining focus group discussions varied from four to 11 and each discussion lasted for 1.5 h to 2 h. Table 1 shows the composition of the focus groups One municipality experienced delays in data collection, leaving nine focus groups for the purpose of this paper.

Topics/questions covered in the FGDs:
Leisure time activities among children and youth;What services and activities exists (in the municipality);Facilitation (who and how) for participation in school, in leisure and at home;Particularly popular/successful initiatives;What should the municipality do to ensure equal participation for the target group;Utilization of parents’ experiences to improve service quality;Identification of improvements needed;Assessment of efforts made by the municipality;Experiences elsewhere that should be considered (in the respective municipalities);Barriers for children and youth with disabilities to participate equally as others;How are municipalities and other actors collaborating to promote participation;Do children and youth with disabilities have a say in service provision;What is participation and what is inclusion (for the target group);Any specific advice to the municipality that can improve service quality.

FGDs were transcribed verbatim and analyzed by following the steps of thematic analysis: familiarization with the data, generation of initial codes, searching for themes, reviewing potential themes, defining and naming themes, and writing their descriptions [22]. Inspired by Anaby et al. [23], we used a deductive approach with elements of ICF as an analytic framework. The data were first categorized into the different ICF environmental factors, then thematically sorted and constructed into themes within the given environmental factors. Finally, the themes were sorted into facilitators, barriers, and requests for improvement.

Two researchers (A5 and A3) organized and coded the data from each of the FGDs. Next, three researchers (A5, A3, and A1) performed a second-order analysis [24] by using the themes from the first analysis to construct themes across the FGDs.

In line with Braun and Clarke [22], we understand qualitative analysis as a situated and interactive process that jointly reflects data, researcher positionality, and research context. Therefore, conducting the analysis as a collaboration between the researchers was seen as a contribution to a deeper reflexive engagement with the data. To ensure the quality of our thematic analysis, we relied on a robust process that included ICF and the concepts of participation and inclusion as our coherent theoretical analytic orientation, and we interpreted the data in light of the FGDs [25].

The development of indicators (Step 4) for participation among youth and children with disabilities drew on different approaches to participation measurement, as shown in Figure 2. Diverse efforts to develop participation scales have been published and we scrutinized a selection of such efforts to distinguish between the different approaches. For instance, the matrix for assessment of activities and participation, based on ICF-CY [11,26,27], asks directly about what children and youth are doing, describing the persons’ actual performances of tasks or actions. The participation scale [28] asks respondents to compare their own participation with their peers. An example of using satisfaction with one’s own participation among adolescents was published by Smits et al. [29]. Finally, we included measures on environmental barriers for participation, such as those that were used in the ICF-CY [11]. The choice of using four different ways of measuring participation (Figure 2) in our study was intended to capture different aspects of perceived participation and, therefore, to establish a more robust and useful instrument. The approach was thoroughly discussed within the research team and with the participating municipalities. This approach was followed by defining relevant items, and questions were formulated with one four-point scale and three five-point Likert scale categories. The themes from the FGDs were scrutinized to identify the formulation of items under each dimension. The dialogue with the participating municipalities was used to finalize the measurement instrument for testing and data collection. 

Table 2 shows the items in the four participation scales (translated from Norwegian and not written in full). 

The questionnaire further included basic demographic information (age, gender, location, education) and a short version of the Washington Group/UNICEF Child Module [30] that is developed for disability screening purposes. Twelve items from the Child Module were used to identify persons who had a lot of difficulty in performing at least one on the 12 activities (Table 3).

The web-based questionnaire was pilot-tested among 90 junior high school children (age range 14 years to 16 years, 57.8% boys) in classroom sessions that were led by one teacher who was instructed by the first-named author of this article. Analyses of pilot data showed high internal consistency (Cronbach’s Alpha between 0.82 and 0.91) and confirmatory factor analyses (CFA) provided scree plots that revealed one main component for each scale and supported construction of four different indices that reflected different dimensions of perceived participation, as illustrated in Figure 2. The oral feedback provided in two of the classes that participated in the pilot test confirmed that the questionnaire was easy to understand and to complete, but the feedback also led to some minor adjustments.

A combined sampling strategy was used for the main data collection. The number of children and youth with functional limitations (age 5 years to 25 years) was estimated to be between hundred and fifty and two hundred in each municipality. Invitations to participate were sent out in two different ways: either (i) the four municipalities sent invitation letters with information about the survey and a link to the web-based questionnaire to all children and youth registered with a disability; or (ii) a control sample with the same number of children and youth in the same age bracket was drawn from the population register and received a similar invitation letter with a link to the questionnaire. A reminder was sent after two weeks. After four weeks, the external agency organizing the data collection submitted the data file to the research team. The study was reported to the Norwegian Center for Research Data.

## 3. Results

### 3.1. Focus Group Discussions

Figure 3 shows cross-cutting themes analyzed against facilitators, barriers, and/or requests. *Family* refers to parents, siblings, grandparents, and other closely related caregivers, i.e., either by consanguinity or affinity. *Technology* is broadly understood as a means to an end, such as communication, knowledge sharing, entertainment, assistance, and mobility. Games, social media, and assistive devices are typical examples of technology. *Volunteering* is understood as time and labor offered freely by individuals or groups engaging in community service. *Leisure activities* refer to various activities, both organized and unorganized, that people engage in during their spare time. *School* is an educational institution. In Norway primary and secondary education usually lasts 13 years. Primary and lower-secondary education (junior high) is a municipal responsibility. *Participation* in decision-making is understood as a process and principle in politics, democracy, and innovation. *Facilitation* includes aspects such as universal design, transitions between service areas, and adaptations of housing, work, and activities. *Services* involve a network of connected elements, such as service culture, employee engagement, service quality, content, comprehensibility, service touchpoints, economy, roles, responsibilities, and communications between stakeholders. The municipalities are responsible for offering a range of services for the target groups; therefore, management and collaboration within and across service areas are important. *Networks* refers to a social structure and to the interactions between individuals or groups such as families with shared interests or experiences. *Attitudes* are understood as established ways of thinking or acting toward someone, such as acceptance and respect of children’s participation. *Communication and information* refer to the mutual exchange of information between municipalities and parents, children, and youth.

Family is identified as a facilitator across municipalities. Family was often mentioned, together with friends, school, or the local community, as an important network resource providing encouragement for participation.

Technology is a facilitator for participation through games and social media that have some obvious positive aspects for children and youth who are less mobile or tend to be less integrated in the “physical” socialization of their peers. Mainstream communication technology, in particular, brings youth into the ongoing flow of communication and sharing in a qualitatively different way than was possible before mobile technology was introduced.

Volunteering and active engagement from individuals are driving forces for social activities in local communities and are described as important pillars of social life, with high value for children and youth with functional difficulties (Service provider, Municipality 3: *Because in our municipality, coming from central management, there is a clear focus on voluntary work and to develop the space for voluntarism*) (All citations are translated from original language). The participation and inclusion of children and youth with functional difficulties often rely on volunteering and a good balance between volunteering and municipal services and activities.

Organized leisure activities were sometimes too demanding and accompanied by rigid rules for participation, leading to the exclusion of some children and youth. The lack of flexibility and competence, or limited resources, were experienced as exclusion mechanisms. On the positive side, well-facilitated activities with good role models for inclusion promoted participation. Good support in activities that were open to everyone provided the children and youth with a feeling of participating. Unorganized leisure activities often played a positive role.

While many parents have positive experiences with primary schools, they sometimes experience a lack of support and a problematic transition into junior high (Parent, Municipality 2: *It was a whole school year with, I am sure, about 10 meetings, where we quarrelled and wept*). Stories were told illustrating that inclusion was not always taken seriously and that the approach could depend on both the individual teacher and the relatively autonomous headmaster. Children may be taken out of class and placed in special classes in junior high, and thereby be excluded from the larger community of school children. There were also reported incidents of unavailable support and/or direct resistance in promoting social inclusion.

Participation in decisions affecting oneself (or one’s children) was sometimes hindered due to an unsystematic approach to collaboration and joint decision-making. Parents mostly did not participate in decision-making that affected their children and youth. This may reflect negative attitudes to parents and children/youth, who were seen by services/service providers as equal partners. Co-determination on matters affecting families directly was requested across FGDs (Service provider, Municipality 3: *And then I would have a user panel (….) both parents and children have a voice that is important for us to consider so that they feel engaged and that they feel they have something to contribute and that there is someone who cares*).

Municipal management and collaboration as barriers had three different elements; general problematic collaboration between services within the municipality; a lack of information and overview across services; and a lack of trust in other’s viewpoints and assessments. From the outside, these barriers resulted in a confusing picture and no clear contact point for inquiries. Requests for improvement mirrored the barriers and included strengthening services (financial and human resources), facilitation of activities, building networks, creating awareness, improving attitudes and behaviors, making information more accessible, and improving the coordination and management of services. Participants requested that municipalities become more proactive in service development, and that parents and children/youth become directly involved in decisions that affect them.

Lack of or sub-optimal facilitation appears as an important barrier for participation among children and youth. This barrier could be in the form of not paying sufficient attention to universal design, simply doing the “wrong” things, or a lack of coordination, internally within the municipality or between municipal services and private initiatives. When facilitation is not carried out or not carried out properly (for instance, when there are limited human resources or physical access is hindered), this constitutes a barrier for participation (Mother, Municipality 1: *If one could have facilitated, I believe they could have participated for a longer time both in handball and football and …. perhaps they could have formed small groups”*). There were, however, some positive experiences of successful inclusion in primary schools, adaptations of housing, and inclusive models for children and youth with disabilities in sports and organized activities (Father, Municipality 3: *He attends the Monday club, a very good offer by the municipality. It is every Monday and he really looks forward to this. They do everything from going to the airport and Burger King, games, cinema, bowling*), and the implementation of universal designs. The facilitation of activities and universal designs were formulated as requests in several of the focus group discussions.

Services may constitute barriers due to limited funding and/or human resources, or inability to cover needs/demands (Service provider, Municipality 3: *We can hardly expect that competence is very high, because there is a father who volunteers, and then there is a mother who volunteers*). Services may be offered too late and/or support for families and children and youth with functional difficulties may be too weak. The lack of low-threshold services was particularly mentioned as limiting the freedom for families and children/youth/young adults. The widely used support contact service often does not live up to the needs and expectations of parents and children/youth, due to high turnover, sub-optimal matching of the support contact and the child/youth, and (in practice) favouring adult contact over peer contact. User-controlled personal assistants are used restrictively by the municipalities, due to costs. Municipalities were regarded as too passive and/or not sufficiently pro-active, unable to adjust, and limited by resources. Finally, sub-optimal external communications, internal systems, and digital tools made it difficult for service users and potential users to obtain a full overview of existing services. On the other hand, services were presented as facilitators when there were actively engaged parents as well as service providers. Relevant and positive support from families, friends, schools, services, and local community were seen as liberating (Parent, Municipality 4: *And the support contact was a young adult.*
*We were very lucky. It was a perfect match*). While the right type of support and the content of services were important, positive support and engagement from service providers and other adults involved with children and youth also appeared to be of high importance for the quality of services. Requests ranged from expanding the user-controlled personal assistant service, providing low-threshold services as well as lowering the threshold in services generally, more resources to strengthen services, and the need for meeting points/arenas.

Limited networks were seen as a barrier, with families losing out on information and support (Mother, Municipality 1: *We do not know any other with the same diagnosis here on the island*). Parents may have no one to contact in a similar situation and can experience being isolated and a feeling of being on the “outside” socially. Municipal services rarely play a role in promoting networks among families of children and youth with functional difficulties, and families struggle to find good “matches” for their children/youth. Building networks and organizing arenas for the exchange of experiences were seen as facilitators for families to find strength in contacts with other families in similar situations (Mother, Municipality 1: *If the public health nurse had invited me and* (name of another mother) *to visit the health center, instead of me having to take contact with her and ask her to my place, then it is better with a neutral meeting place where I have not taken the initiative but the public health nurse*).

Negative attitudes and behaviors and the use of stigmatizing/negative language often hindered active social participation. This may have been unintentional, but can have dire negative impact. In contrast, positive attitudes were mentioned as important facilitators. Accepting children “as they are” and using positive language may lead children and youth with functional difficulties to feel proud of themselves, which promotes participation and reduces exclusion and isolation. An example was the word “functional variation” that was coined by one of the youth participants as an alternative to the more stigmatizing word “disabled” or similar words. Children and youth requested more attention to language and respect; parents requested support that provides their children with the freedom to make their own choices; and service providers wanted to strengthen families.

Limited communication and information between the municipality and parents was experienced by parents, adding to the feeling of being isolated. Many parents did not know where to go if they needed to communicate with the municipality. An understanding of the importance of good communication can push municipalities to adopt a more conscious communication strategy.

### 3.2. Survey

Correlational analyses confirmed, as expected, that the correlation between environmental barriers and the WG/UNICEF disability scale was positive (Pearson: 0.44), while both were negatively correlated with comparing with others/peers and one’s own participation (−0.42 and −0.65). Correlations between one’s own participation, satisfaction with one’s own participation, and comparing one’s own participation with that of others/peers were, however, mixed; i.e., they were significantly and non-significantly positive. Of the 186 respondents, 84 were persons with disabilities within the current operational definition (with considerable difficulty on one question). Before adding the items to form scales, missing values (varying from 10 to 30 missing values) were replaced by a mean. Table 4 shows the scale properties and the analyses comparing persons with and without disabilities along the four scales, revealing that persons with disabilities reported lower levels of their own participation, assessed their own participation as lower than that of their peers, and were less satisfied with their own participation. Environmental barriers were, as expected, higher for persons with disabilities than for persons without disabilities.

## 4. Discussion

The high-level focus group discussion analyses identified facilitation and universal design, pro-active municipal services, good networks, positive attitudes on inclusion, good communication between municipalities and families, schools, and inclusive leisure activities as critical factors in ensuring that children and youth with functional difficulties are included and participate in different arenas in their local communities. Whether each of these factors appeared as a barrier or a facilitator for participation depended on the specific context and may have reflected the level of resources as well as general awareness about the rights of children and youth with functional difficulties. Supporting families, technology as vehicle for communication, and volunteering were presented as facilitators, while internal communication and organization within municipalities, lack of co-determination, and school environments where support was lacking were chiefly reported as barriers. While each of the themes point to critical areas for ensuring inclusion and participation, it is important that there be a mutual influence among the different areas. For instance, an explicit commitment by the municipalities to the rights of children and youth with functional difficulties will influence resource allocations and stimulate services positively. An active voluntary sector in a municipality will not only ensure a broad spectrum of activities, but also influence priorities at the municipal level. Active and resourceful parents can apply pressure on a municipality with respect to both resource allocation and service content. Collaboration between schools, municipalities, and parents may be positively influenced by inclusive school policies and practices. Well-organized municipal services and internal coordination of those services will improve access to the services.

The indicators developed in this project are new, adapted to the target groups and the context, reflect the purpose of the research and the innovations, and represent different aspects of self-perceived participation. The statistical analyses provided support for establishing four different scales. The results indicated that all four scales demonstrated an expected difference between children and youth with and without disabilities, providing support for the survey’s content validity. Although further refinement may be necessary (for instance, to better capture the perception of barriers to participation), the four scales appeared to provide a solid method of measuring perceived participation among children and youth in the study settings. Previous attempts to measure participation have combined the frequency and level of involvement in activities [14,16,17], or applied comparison with peers [28] or satisfaction measures [29], often targeting specific impairments. Therefore, our study’s combination of different approaches in measuring perceived participation adds to the literature on participation measures. Further testing, validation, and planned repeated data collection by the municipalities may lead to a refinement of the scales. The current study’s new approach in measuring participation is focused on using different aspects of perceived participation, rather than relying on one or more aspects across different domains. While the ICF model and its definition of participation influenced this research on a general conceptual level, our approach has developed a contextually adapted instrument that can be useful for participating municipalities when they develop policies and interventions to promote participation, rather than a general and “context-free” instrument that would be less informative for interventions in specific contexts. Therefore, our approach may contribute to a more multi-facetted concept of participation in the future development of participation measures and theoretical models for participation.

The results of the survey, in combination with the focus group discussions, may contribute to a better understanding of barriers and facilitators for participation and provide a benchmark for existing quality indicators. It is expected that the current results and more in-depth analyses of both the FGDs and the survey data will motivate municipalities to scrutinize their services to find areas of improvement and/or to identify services that need to be revised or re-evaluated to shift priorities and change services that are aimed at inclusive communities. The upcoming service design process involving municipalities, parents, and children/youth will aim to challenge municipalities to better adapt their services and to organize the services to the needs of the target groups. Not least, is it expected that attention to inclusion and participation via comprehensive research and innovation collaboration will positively influence awareness about the rights of children and youth who are in need of support.

The presentation and discussion of the combined methods used for the current study are demanding in an article format, which have impacted the depth of the methods that were used. In this case, the presentation of the FGDs could have been more detailed, and could have included the first level of the analyses. While the sample for the survey was adequate for the purpose of comparing between persons with and without disabilities, it is more problematic with respect to the representativity of children and youth in the participating municipalities. The results of comparing between persons with and without disabilities indicated that the scales are valid, but further refinement and testing will be necessary to produce a final version.

## 5. Conclusions

The study considered the measurement of participation and inclusion of disabled children and youth by combining four different approaches to perceived participation. We implemented a comprehensive research process by combining research methods and a participatory approach in which four municipalities were closely involved throughout the study, as well as by ensuring that the voices and input of children, youth, and parents were considered. To reach the aim of promoting the participation of children and youth in daily life situations within the context of the selected municipalities’ responsibilities, it was critical to establish a collaborative foundation of committed, active, and engaged municipalities, as well as parents, children, and youth. It is easy for researchers to underestimate the time and resources that are needed to establish and maintain this foundation in practice. To maintain mutual respect in the collaborative study, it was necessary for us to understand that we were competing for attention and time in the context of rather busy schedules and everyday challenges for all of the participants. Patience and long-term commitment are keys in increasing awareness, influencing priorities, and establishing practical and collaborative arrangements. Innovative methods were necessary in breaking through the glass ceiling of repeated studies of barriers that led to marginal changes. A successful implementation of this research and innovation may subsequently inspire other municipalities to engage in similar processes.

## Figures and Tables

**Figure 1 ijerph-19-11893-f001:**
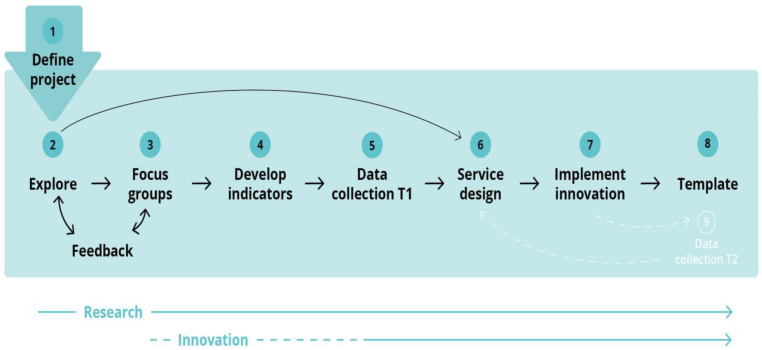
Research and innovation cycle in this project.

**Figure 2 ijerph-19-11893-f002:**
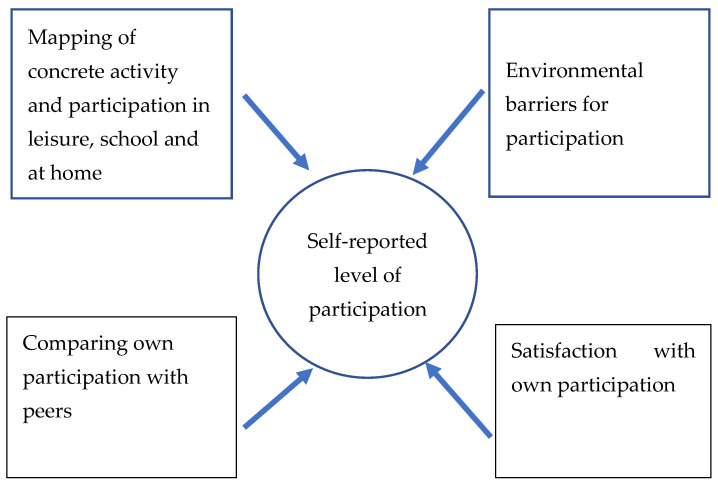
Participation indicators in the questionnaire.

**Figure 3 ijerph-19-11893-f003:**
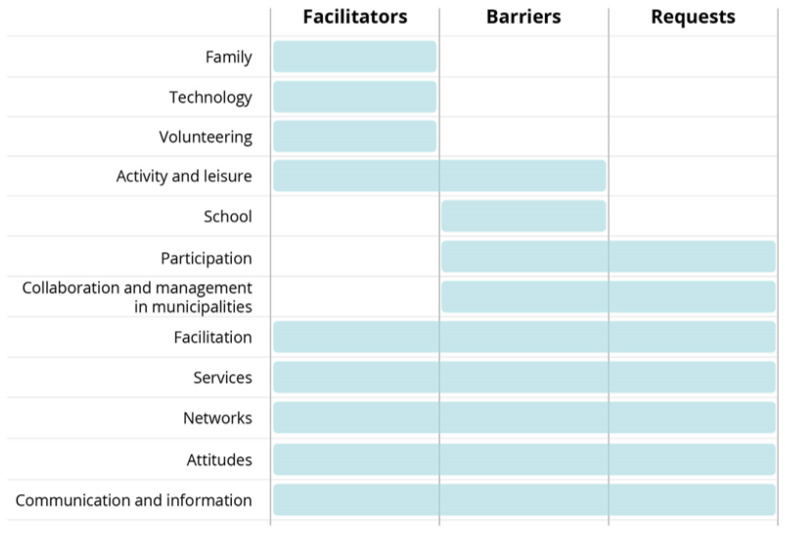
High-level Analyses of Cross-Cutting Themes (three municipalities).

**Table 1 ijerph-19-11893-t001:** Participants in Focus Group Discussions.

Municipality	Service Providers (Male/Female)	Parents (Male/Female)	Children		
			Boys	Girls	Age Range
A	8 (3/5)	6 (3/3)	1	4	19–23 (34) ^1^
V	6 (2/4)	5 (1/4)	3	2	10–20
H	11 (2/9)	3 (0/3)	1	3	18–25
S	8 (1/7)	3 (0/3)	1	4	13–28 (30) ^1^

^1^ Criteria for selection of children and youth were ages of 12 to 25 years, withability to contribute in an FGD. Three of the participants were above 25 years of age, but assessed by the authors as suitable for the purpose of the study.

**Table 2 ijerph-19-11893-t002:** Participation scale items.

Level of Own Participation
Children and youth participate in many different activities during a month. These activities can take place at home, at school, or during leisure periods—both organized and unorganized activities (1–5: never, seldom, sometimes, often, very often). How often are you: -carrying out leisure activities at school -together with friends -carrying out leisure activities (outside school with friends) -performing organized sports -visiting a cafè together with others -in contact with others on social media -performing leisure activities with the family (other family members) -attending cultural events with someone other than family members -visiting friends -talking to others that are not in your family -friends visiting you
Environmental barriers
Do you experience barriers for participating in activities at home, at school/work or leisure? (1–4: to a large degree, to some degree, to a small degree, not at all): -difficult to get transport to go to the activities -too expensive to participate -lack support to participate -negative attitudes in the society -activities are not adapted to my needs -lack of information from the municipality -activities for children and youth are not prioritised -accessibility is problematic -lack overview of activities -difficult to get the right assistive products
Satisfaction with own participation
How satisfied are you with your own participation when you are (1–5: very satisfied, satisfied, neither nor, dissatisfied, very dissatisfied): -together with friends at your place (in your own home) -doing leisure activities other than sports activities with friends -visiting friends (in their home) -in contact with others on social media -at school/work -doing leisure activities with your family -at cultural events with friends or other than your family -doing organized sports activities
Own participation compared to others
Compared with people your own age, how often are you (1–5: never, seldom, sometimes, often, very often): -together with friends (outside your own home and their home) -doing leisure activities (other than sports) -doing organized sport -at meetings/arrangements in a voluntary organization (other than sports) -visiting a café with others than your family -in contact with others on social media -at leisure activities with your family -at cultural events with friends or other than your family -visiting friends (in their home) -talk to other than family members -doing leisure activities at school -together with friends at your place (in your own home)

**Table 3 ijerph-19-11893-t003:** Washington Group/UNICEF Child Module items.

Washington Group/Unicef Child Module Items Used in the Survey
Do you have any difficulties with activities in daily life related to your health? (1–4: No difficulties, some difficulties, a lot of difficulties, unable to do (the activity): -seeing, even if you are wearing glasses -hearing, even if you are using hearing aid -walking or climbing stairs -remembering or concentrating -with self-care, such as washing all over or dressing -using your own language, have difficulty communicating, for example understanding or being understood -learning things (compared to children your own age) -remembering (compared to children your own age) -concentrating on something you like to do -accepting changes or routines -controlling yourself (compared to children your own age) -making friends

**Table 4 ijerph-19-11893-t004:** Scale properties and level of participation among persons with and without disabilities (*n* = 186).

	Scale Properties	Mean	Standard Deviation	KMO Bartlett’s	*p*	% Variance
						1st Factor
Scale (Range) ^1^		Factor				
Level of own participation (20-65)		41.35	8.11	0.81	<0.001	20.4
Environmental barriers for participation (12–48)		25.10	7.87	0.91	<0.001	30.1
Own participation compared with peers (12–48)		31.07	7.32	0.85	<0.001	31.9
Satisfaction with own participation (11–55)		41.09	5.80	0.88	<0.001	31.6
		Mean scale value			F	*p*
		Persons with disabilities		Persons without disabilities	9.83	<0.01
Level of own participation (20-65)		40.25		42.68	4.22	<0.05
Environmental barriers for participation (12–48)		26.68		23.18	9.55	0.002
Own participation compared with peers (12–48)		29.53		32.95	10.61	0.001
Satisfaction with own participation (11–55)		39.83		42.62	11.27	<0.001

^1^ All four scales show increasing levels of participation, environmental barriers, and one’s own participation and satisfaction, with increasing scale values.

## Data Availability

The research project is ongoing and data have not yet been publicly archived. Data are available from the first author upon request until archived at SIKT.

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
