# Peer review of "Participation and Inclusion of Children and Youth with Disabilities in Local Communities"

_ijerph, 2022, doi:10.3390/ijerph191911893_

Round 1

Reviewer 1 Report (Previous Reviewer 2)

This revision is a great improvement over the original version, addressing my main concerns. The one major area which I believe still needs work is the discussion:

·       In lines 510-12, the authors state: “The combination of different approaches to measuring perceived participation in one study adds to the literature on participation measures….” They need to relate the findings to specific literature in the area to show more clearly how the results moves research forward. They can refer back to work cited in lines 85-90 of the study.

·       No mention is made of study limitations in the discussion. These limitations need to be identified.

·       In lines 512-13, they mention that the study “may be further developed through including other aspects such as for instance the importance of participation.” This statement is insufficient for identifying the research’s next steps, i.e., the authors need to flesh out future directions for inquiry more specifically.

Other comments are that the headings in Table 1 of the results (page 11) still do not match the discussion of the results on pages 11-13. Also, there are two Table 1s—one on pages 7-8 and one on page 11.

Finally, the paper needs to be thoroughly proofread as there are numerous grammatical mistakes.

Author Response

Reviewer 1:

Lines 510-512: A sentence has been included ahead of the statement (about adding to the literature), referring to literature form the introduction and the methods section, providing further details to support that this study adds to the literature on measuring participation. 

Study limitations have been addressed in a separate paragraph at the end of the discussion

Lines 512-13: The plan is to repeat data collections (survey) one or more times in the near future. This is however not decided (meaning: funded). We have chosen to reformulate to avoid anticipating inclusion of additional aspects.

Table 1 – about mismatch between headings in the table and the discussion. This has been corrected. The themes are listed in the left column in the table. Facilitators, barriers and requests constitute  the framework for the analyses as explained earlier in the text.  

Table numbering has been corrected.

Language has been edited.

Reviewer 2 Report (New Reviewer)

The authors present a well conceived manuscript describing services and opinions from Norway, a country that offers community services for children and youth with disabilities. They utilized both focus groups and surveys, that were pilot tested, to better understand facilitators, barriers and requests for services. Overall it is well written and offers unique insights to the field about facilitating community participation for children and youth with disabilities. The manuscript will benefit from English language editing. 

Author Response

Reviewer 2:

The language has been edited

This manuscript is a resubmission of an earlier submission. The following is a list of the peer review reports and author responses from that submission.

Round 1

Reviewer 1 Report

This is a otentially important research project. However, I cannot recommend publication in its present form. Suggestions are given below

1. The introduction is acceptable

2. From that point on there are insufficient details on methodology and results  In order to evaluate the adequaqcy of the study. Methods should be divided into Methodology 1. Including construction of the focus groups and ideally a table of participants in each group. In additionmore details are needed on analysis of the results (line 146), including who carried out categorisation, thematic analysis etc and the reliability of these. Quotes from focus groups, possibly divided into child, parents service providers, would help to exemplify the 13? areas highlighted in the results 2. The construction of the questionnaire was not described in  detail, and a specimen should be provided . We are not told how many questions, grouped into areas or not, nor what measurement scale was used.

Results Pages192-198 could go in the Method section. Table 1 appears out of the blue; all themeasures on the L hand side need to be defined and specific examples given in the text. The subsequent list of issues could be organised under the heading of Facilitators, Barriers , Requests..

The questionnaire should already have been introduced in Method. Results should be included (page 322).

P330 From here on these are not results, but rather what is to be done in the future ie. Discusssion. Putting them in Results is confusing. 

Discussion P420 should be in the results section

In summary a worthy piece of research, let down by the write up. The article would benefit from a clearer structure, more details of key processes and less repetition.

Author Response

Response to reviewer 1 point by point

Reviewer 1

Comments and Suggestions for Authors

This is a otentially important research project. However, I cannot recommend publication in its present form. Suggestions are given below

  1. The introduction is acceptable
  2. From that point on there are insufficient details on methodology and results  In order to evaluate the adequaqcy of the study. Methods should be divided into Methodology 1. Including construction of the focus groups and ideally a table of participants in each group. In additionmore details are needed on analysis of the results (line 146), including who carried out categorisation, thematic analysis etc and the reliability of these. Quotes from focus groups, possibly divided into child, parents service providers, would help to exemplify the 13? areas highlighted in the results 2. The construction of the questionnaire was not described in  detail, and a specimen should be provided . We are not told how many questions, grouped into areas or not, nor what measurement scale was used.

Methods and results have been reorganised and expanded. We have included more details about the analysis including example quotes from the FGDs as examples

Results Pages192-198 could go in the Method section. Tab le 1 appears out of the blue; all themeasures on the L hand side need to be defined and specific examples given in the text. The subsequent list of issues could be organised under the heading of Facilitators, Barriers , Requests..

An explanation to Table 1 has been included prior to the table. Methods and Results have been reorganised to meet the comment. We have further chosen to organise the list of issue (after Table 1) according to the order of themes.

The questionnaire should already have been introduced in Method. Results should be included (page 322).

More details on the questionnaire are included in the text and we have added a table with the key indicators. Note that the questionnaire is in Norwegian language, so we have translated the key indicators and chosen not to attach the questionnaire as a supplementary file.

P330 From here on these are not results, but rather what is to be done in the future ie. Discusssion. Putting them in Results is confusing. 

The text on service design has largely been omitted and steps 6 – 8 are now only mentioned as part of the overall research process. These are steps that will be implemented later (project is ongoing).

Discussion P420 should be in the results section

We have reorganised methods, results and discussion

In summary a worthy piece of research, let down by the write up. The article would benefit from a clearer structure, more details of key processes and less repetition.

Reviewer 2 Report

While this study is a notable step forward and has the potential to be an excellent piece of research, I do not believe it is ready for publication in its current form. Most importantly, the results are incomplete without presenting the findings of the quantitative survey mentioned as a next step. One needs these quantitative results to make conclusions which would be meaningful to practitioners and researchers. Second, it is unclear how this approach advances the literature on participation/inclusion of children and youth with disabilities. There is no mention of work in this area in other localities, and how the presented approach moves the needle forward in this research area. Third, there is no grounding of the approach in any theory so that the study does not provide any benefit to the theoretical literature on the subject matter. Once the authors complete the quantitative study, and address the second and third points mentioned above, they could resubmit the paper.

Additional comments are as follows:

Introduction

·       The authors need to define exactly what they mean by “children” and “youth” (first mentioned in line 26).

Method

·       All of a sudden, the authors talk about “children, youth and young adolescents” (line 143) rather than just “children and youth”, which is very confusing.

·       When first introducing the term, “co-design” (line 152), it should be defined.

·       There should be a table which defines the number of people, gender composition, and age distribution of the each of the nine focus groups.

Results

·       In Table 1, the wording of three themes should be changed to better match the discussion: “Leisure activities” (rather than “Activity and leisure”), “Participation in decision-making” (rather than “Participation”), and “Municipal Management & Collaboration” (rather than “Collaboration and management in municipalities”).

·       When presenting the results for each of the themes, sub-headings should be used, i.e., sub-headings for “Facilitation”, “Services”, etc. The addition of these sub-headings will make the section easier for the reader to follow.

·       There is no discussion of the “family” theme.

Indicators of Participation

·       The authors should explain how each of the three different sources for developing the participation measure influenced the development of the scale. Presently, they just make the general statement that these sources affected scale development without providing specific detail.

·       “v” should be changed to “iv” in line 317.

·       Data for the Cronbach’s Alpha and Confirmatory Factor Analyses mentioned in lines 325-329 should be provided.

·       All of a sudden, the authors add “young adults” (line 334) rather than just “children and youth”, which is very confusing.

·       The future data analyses mentioned in the paragraph starting on line 339 should also include a comparison of the test and control samples.

Author Response

Reviewer 2

Comments and Suggestions for Authors

While this study is a notable step forward and has the potential to be an excellent piece of research, I do not believe it is ready for publication in its current form. Most importantly, the results are incomplete without presenting the findings of the quantitative survey mentioned as a next step. One needs these quantitative results to make conclusions which would be meaningful to practitioners and researchers. Second, it is unclear how this approach advances the literature on participation/inclusion of children and youth with disabilities. There is no mention of work in this area in other localities, and how the presented approach moves the needle forward in this research area. Third, there is no grounding of the approach in any theory so that the study does not provide any benefit to the theoretical literature on the subject matter. Once the authors complete the quantitative study, and address the second and third points mentioned above, they could resubmit the paper.

The text on development of indicators has been expanded, including details on scale construction and results of analysing differences on the key indicators between children and youth with and without disabilities. References to other relevant studies have been included in the introduction and an argumentation for the contribution of our contribution in the discussion. The conceptual link to ICF is included and implications for theory on participation included in the discussion.

Additional comments are as follows:

Introduction

  • The authors need to define exactly what they mean by “children” and “youth” (first mentioned in line 26).

                  A definition has been included in a footnote

Method

  • All of a sudden, the authors talk about “children, youth and young adolescents” (line 143) rather than just “children and youth”, which is very confusing.

This has been changed.- We are now using "children and youth" throughout the text

  • When first introducing the term, “co-design” (line 152), it should be defined.

The section on co-design has been removed. The last three steps on the overall project will be implemented later (2023) and are now only mentioned in the article. After considering we decided to use "service design" for clarity and have included a definition in a footnote. 

  • There should be a table which defines the number of people, gender composition, and age distribution of the each of the nine focus groups.

                        A table has been included as requested

Results

  • In Table 1, the wording of three themes should be changed to better match the discussion: “Leisure activities” (rather than “Activity and leisure”), “Participation in decision-making” (rather than “Participation”), and “Municipal Management & Collaboration” (rather than “Collaboration and management in municipalities”).

We have changed the wording of the three themes according to the comment

  • When presenting the results for each of the themes, sub-headings should be used, i.e., sub-headings for “Facilitation”, “Services”, etc. The addition of these sub-headings will make the section easier for the reader to follow.

We have re-organised the presentation of results so that it is ordered in the same way as the      cross-cutting themes. This should make it easier to read and align with Table 1.

  • There is no discussion of the “family” theme.

A text about the "family" theme has been included

Indicators of Participation

  • The authors should explain how each of the three different sources for developing the participation measure influenced the development of the scale. Presently, they just make the general statement that these sources affected scale development without providing specific detail.

The explanation of construction of the participation measure(s) has been revised and expanded  

  • “v” should be changed to “iv” in line 317.

This part of the text has been moved and revised

  • Data for the Cronbach’s Alpha and Confirmatory Factor Analyses mentioned in lines 325-329 should be provided.

Chronbach's Alphas have been included in the revised text

  • All of a sudden, the authors add “young adults” (line 334) rather than just “children and youth”, which is very confusing.

The revised text uses "children and youth" throughout

  • The future data analyses mentioned in the paragraph starting on line 339 should also include a comparison of the test and control samples.

This text has been revised and the new version includes analyses of the main data. We have also added some more information about the test sample.